# Detection and Characterization of Methylated Circulating Tumor DNA in Gastric Cancer

**DOI:** 10.3390/ijms25137377

**Published:** 2024-07-05

**Authors:** Seung Young Seo, Sang Hee Youn, Jin-Han Bae, Sung-Hun Lee, Sun Young Lee

**Affiliations:** 1Department of Internal Medicine, Jeonbuk National University Medical School, Jeonju-si 54907, Republic of Korea; 2Research Institute of Clinical Medicine of Jeonbuk National University-Biomedical Research Institute, Jeonbuk National University Hospital, 634-18 Keuman-dong, Dukjin-gu, Jeonju-si 54907, Republic of Korea; younsanghee@gmail.com; 3Department of Radiation Oncology, Jeonbuk National University Medical School, Jeonju-si 54907, Republic of Korea; 4Research Center, Cancer Breaker, Yongin-si 16942, Republic of Korea; 82jinhan@gmail.com; 5Cancer Genomic Research Institute, Clinomics, Chungju-si 28161, Republic of Korea; shlee@clinomics.co.kr

**Keywords:** gastric cancer, liquid biopsy, MeDIP, fragment

## Abstract

Gastric cancer is the fifth most common disease in the world and the fourth most common cause of death. It is diagnosed through esophagogastroduodenoscopy with biopsy; however, there are limitations in finding lesions in the early stages. Recently, research has been actively conducted to use liquid biopsy to diagnose various cancers, including gastric cancer. Various substances derived from cancer are reflected in the blood. By analyzing these substances, it was expected that not only the presence or absence of cancer but also the type of cancer can be diagnosed. However, the amount of these substances is extremely small, and even these have various variables depending on the characteristics of the individual or the characteristics of the cancer. To overcome these, we collected methylated DNA fragments using MeDIP and compared them with normal plasma to characterize gastric cancer tissue or patients’ plasma. We attempted to diagnose gastric cancer using the characteristics of cancer reflected in the blood through the cancer tissue and patients’ plasma. As a result, we confirmed that the consistency of common methylated fragments between tissue and plasma was approximately 41.2% and we found the possibility of diagnosing and characterizing cancer using the characteristics of the fragments through SFR and 5′end-motif analysis.

## 1. Introduction

### 1.1. Gastric Cancer and Liquid Biopsy

Despite advances in diagnosis and treatment, cancer is the second leading cause of death worldwide [1]. In particular, gastric cancer is the fifth most common disease and fourth most cause of death, causing 1.1 million new cases and 7.6 hundred thousand deaths each year worldwide [2,3]. Recently, the incidence of gastric cancer has been increasing in Korea due to changes in eating habits and various environments [4,5]. The gold standard of diagnosis for gastric cancer is esophagogastroduodenoscopy with biopsy. However, the diagnosis can often be delayed in early-stage cancer. In particular, liquid biopsy research using blood has been conducted recently for early diagnosis of various diseases, including cancer [6,7]. In various diseases, including cancer, various characteristics derived from the lesion are reflected in the blood. Direct indicators include exosomes, metabolic materials, cfDNAs, and CTCs. Although it exists in extremely small quantities, analyzing its characteristics has made it possible to diagnose the disease with minimally invasive methods [8,9,10]. Therefore, for a more accurate diagnosis, the unique characteristics of each cancer type must be identified. Representative successful cases include EGFR, KRAS, and BRAF mutations in lung cancer [11,12,13,14]. Recently, omics research on various characteristics has reached a level where it is possible to diagnose not only cancer but also types of cancer [15,16,17]. However, the uniqueness of many types of cancer is still unclear.

### 1.2. Methylation

Changes in the gene sequence cause functional changes and, in fact, many diseases, including cancer, are caused by small changes in genes [18]. It is known that cancer is sometimes caused by abnormal changes in expression levels rather than functional changes in genes. These epigenetic studies are widely applied in the diagnosis and prediction of cancer. Human cancer is closely associated with changes in methylation [19]. In particular, tissues sometimes have specific methylation profiles [20,21]. Based on these characteristics, it is possible to diagnose cancer and its type using the methylation pattern of cancer-derived substances present in the blood.

### 1.3. Fragmentomics (Size and 5′End-Motif)

cfDNA in the blood comes from various tissues and cells [22,23]. Recently, it has been revealed that fragments derived from various cancers are shorter than those derived from normal cells due to different biological mechanisms, and the possibility of diagnosis using this is being discussed [24,25,26,27].

All living organisms contain nucleases that can interact with nucleic acids and hydrolyze phosphodiester bonds. Nucleases may also exhibit a cleavage preference for single- or double-stranded nucleic acids. Some nucleases are structure- or sequence-specific. In particular, there are nucleases that are predominant and specific to the cancer type or tissue [28,29,30,31]. In gastric cancer, FEN1, APE1, XPF/XPG, MRN complex, and DNase1 are typically found [30].

Many liquid biopsy studies are focused on diagnosing cancer by analyzing the characteristics of cancer-derived cell-free nucleic acids in the blood. However, it is still true that in the early stages of cancer, the amount present is very small, and a lot of effort and time are required for accurate analysis. In this study, in order to clarify the characteristics of cancer-derived fragments, hypermethylated fragments were collected and their length and 5′end-motif sequences were analyzed. We also confirmed whether this approach was consistent with results from tissue.

## 2. Results

### 2.1. Differentially Methylated Region (DMR)

First, changes in cfDNA methylation in the plasma of cancer patients were compared with the healthy human plasma. Paired blood and tissue from 22 stomach cancer patients and 40 plasma samples from healthy people were used (Table 1). The MEDIPS library was used to determine differentially methylated regions (DMR) [32]. The BH (Benjamini–Hochberg) method was set to be used for *p*-value adjustment, and areas with a value of 0.01 or less based on the adjusted *p*-value were selected as DMR. The methylation marker information for each cancer type was used as published in Vrba’s paper [33], and the methylation marker information for each tissue was used as published in Moss’ paper [34]. A total of 1,541,364 DMRs were obtained by comparing tissue and plasma of cancer patients with healthy human plasma, and the total length was approximately 813 Mb (Figure 1A and Appendix A). About 146 Mb DMRs were identified between plasma from cancer patients and healthy human plasma, with a ratio of 124 Mb gain (hypermethylation) to 22 Mb loss (hypomethylation) of approximately 6:1. Additionally, DMR coverage of 667 Mb was confirmed between cancer patients’ tissue and healthy human plasma, and hypermethylation and hypomethylation of 528 Mb and 138 Mb were found, respectively (Figure 1). Between cancer patient and healthy plasma samples, approximately 3% (9711/316,507) of DMRs were found in the promoter region. Between cancer tissue and healthy plasma samples, 2.6% (31,922/1,224,857) of DMRs were located in promoter regions (Appendix A and Appendix A).

### 2.2. Commonly Methylated Region (CMR)

Next, we analyzed how much methylated DNA there was in common between tissue and plasma from cancer patients. The MACS-2.2.6 program was used to discover methylation regions [35]. The discovered methylation region was then used to determine the common methylation region (CMR). Methylated regions were processed in 300 bp sections. If the methylated region discovered by the MACS program occupied more than 50% of the 300 bp region, the region was considered methylated. The region was determined by connecting adjacent windows (Appendix A). Overall, the average number of peaks was 439,109 and the average number of CMRs was 213,548, and most CMRs were not located in DMR (Appendix A). When comparing the tissue and plasma of cancer patients with that of healthy human plasma, the average size of the CMR between tissue and cfDNA was approximately 73 Mb (Figure 2). Interestingly, the average similarity was approximately 41.2% between tissue and cfDNA (Appendix A). This shows that cancer-tissue-based methylation changes were also reflected at high levels in ctDNA from the blood.

### 2.3. Short Fragment Ratio (SFR)

To detect cancer, we used cfDNA obtained from MeDIP to compare the length of cfDNA in the plasma of cancer patients and the plasma of healthy donors. Many previous studies have already shown that shorter-length cfDNA exists in patients with various diseases, including cancer. Although ctDNA derived from cancer is only a small portion of the total, we distinguished the difference more clearly by using the proportion of short cfDNA. The insert size estimated from the mapping information was used as the length of cfMeDNA (circulating free methylated DNA). The short fragment rate (SFR) was calculated as the proportion of short fragments among all fragments. cfMeDNAs with a length of 100 to 240 bp were set as full fragments, and those with a length of 100 to 160 bp were set as short fragments. To calculate the skewness of the cfMeDNA length distribution, 100 to 240 bp fragments were used. Simply looking at the trend, the proportion of short cfDNA in the patient’s plasma was high (Figure 3A,B). Next, for detailed quantification, we calculated the SFR by dividing healthy samples and cancer patients and compared the differences. In conclusion, a high level of significant SFR difference was observed in cancer patients (*p* value = 1.63 × 10^−8^; Figure 3C). Surprisingly, focusing on the CpG island resulted in a clearer distinction (*p* value = 1.97 × 10^−11^; Figure 3D). To verify the ability to distinguish between normal and cancer samples using SFR, the ROC curve was calculated. At the whole genome level, the AUC was 0.909. Interestingly, applying this to the CpG island region raised the AUC to 0.950 (Figure 3E,F). We identified how SFRs differed in the promoter regions of 51 well-known cancer-related genes [36]. In particular, the SFR of the BAP1 gene promoter was statistically significantly higher in cancer patients (Figure 4).

### 2.4. 5′End-Motif

Short fragments occurred depending on the type of acting DNase. In particular, it is known to show characteristic differences when DNA is fragmented within cancer cells. We classified the collected short cfDNAs using the sequences of their 5′end-motifs. The length of the motif used in the 5′end-motif frequency analysis was 4 bp. There were 256 possible 4-mer motifs, among which 199 motifs showed significant differences between gastric cancer patients and healthy subjects. Here, 115 and 83 motifs were frequent and infrequent in gastric cancer patients and healthy people, respectively. In other previous studies, abnormal activation of DNase I was reported in gastric cancer [30]. As a result of the action of DNase I, a thiamine end-motif was generated, and in our results, T-end SFRs accounted for a statistically significantly high proportion in the top 10 rank (Table 2). Usually, when cfDNA is generate, the A-end sequences are generated by DFFB (DNA fragmentation factor B). Interestingly, in our significant list, T-end motifs were frequent in gastric cancer patients, and A-end motifs were frequent in healthy subjects. The Mann–Whitney U test was used as the statistical test for the frequency of differences between groups, and the list of significant motifs was calculated based on the *p*-values.

## 3. Discussion

We hypothesized that there would be a clearer difference between cancer and healthy samples if methylation changes were limited to cfDNA. In this study, hypermethylated cfDNA predominated in both cancer and healthy donor plasma due to MeDIP capture. Hypermethylation was more prevalent in cancer genomes. As our results showed, MeDIP had the effect of making these characteristics more evident. It is well known that many diseases, including cancer, are related to methylation change. Moreover, methylation change is related to age-, gender-, species-, and organ-specific factors [37]. However, to be free from these factors, individual cancer-derived substances were compared with a pool of 40 healthy donor plasma results.

Of course, comparing within the same type of specimen can yield more accurate results than comparing plasma and tissue. Nevertheless, there are many studies attempting to determine whether the characteristics of cancer tissue are reflected in blood for liquid biopsy. We obtained common DMRs between cancer tissue and plasma of cancer patients compared with plasma of healthy people. Although a more in-depth study of the obtained results was not conducted, we believe it will be good material for future research.

Because methylation differences were well reflected using MeDIP, we expected to discover CMRs within the DMRs. An extreme DMR was also set between cancer and normal plasma for dramatic results. Unfortunately, the CMR seems to have little to do with the DMR. This result is expected to have occurred because a very small amount of cfDNA, including ctDNA, was used as a control. Nonetheless, surprisingly, there was a match rate of about 40% between cancer tissue and cancer patient plasma. This was the result of the characteristics of cancer tissue being reflected in the blood. Sometimes, many tissues have tissue-specific mutations and methylation patterns depending on the type of cancer [38,39]. It is expected that the CMR may contain clues that can predict not only the presence or absence of cancer but also the type of cancer.

Short-fragment DNA provides useful clues for ctDNA classification using liquid biopsy. Especially in the early stages, ctDNA accounts for less than 1% of cfDNA [40,41]. Therefore, there are limitations in clearly distinguishing cancer from normal samples with only these small changes. We hypothesized that cfDNA derived from cancer could be more clearly identified through methylation capture. Surprisingly, when our method used SFR to distinguish cancer from normal samples, the AUC of the ROC was 0.909. Additionally, when limited to CpG islands, the AUC increased to 0.95. This showed that SFR can more clearly distinguish between cancer and normal conditions under certain conditions, such as methylation. We applied methylation capture and CpG island piecemeal, but we expect that better discrimination ability will be achieved in the future.

All living organisms have enzymes that break down nucleic acids, and DNases and RNases act differently depending on the type of tissue or disease. The action of these different DNases resulted in shorter lengths of cancer-derived cfDNA. In particular, all nucleic acids exposed due to apoptosis or necrosis were fragmented by various DNases and RNases, and some were exuded into the blood. Verification is necessary to clearly reveal the direct basis for the relationship between DNase and 5′end-motif in the results presented in our study. Nevertheless, our study showed that cancer diagnosis is possible by utilizing the characteristics of ctDNA in cfDNA. In particular, if the DNase action is more clearly identified for each cancer type, it is expected that cancer can be diagnosed using only the short sequence of the fragment (only 4 bp).

Although the genes we used in our analysis were limited to 51, these genes are already well-known cancer-related genes. Genes with high methylated-SFR in cancer patients, such as *AKT1*, *BAP1*, *BRCA1/2*, *BRAF*, *CCND1*, *ERBB2*, *ESR1*, *FGFR1/2/4*, *FLT3*, *HRAS*, *IDH1*, *KRAS*, *MAP2K1*, *PPARG*, *PTCH1*, *PTEN*, *ROS1*, *SMO*, and *TP53*, may act as direct or indirect suppressors in gastric cancer. In particular, *BAP1* is a well-known TSG, and its role in gastric cancer has also been identified. This study did not conduct experiments to support the function of highly frequent, methylated SFR genes, including *BAP1*, in gastric cancer. However, shorter fragmentation and methylation of TSGs and oncogenes may not be unrelated to the development of cancer. Characteristics of the fragments, such as differences in methylation or the length of these genes in the blood, may be the result of their origin from actual cancer tissue. In other words, identification of these characteristics can provide a clear basis for cancer diagnosis using liquid biopsy. Unfortunately, gastric cancer was not ranked in the classification using tissue-specific hypermethylation markers. Perhaps methylation capture may simplify features of the tissue, including stomach, that we were unaware of.

In conclusion, we initiated this study out of curiosity to see whether known properties of ctDNA could be further elucidated through methylation. Although there were limitations in obtaining normal controls, we believe that our approach offers new possibilities for future gastric cancer diagnosis and characterization using liquid biopsy. We plan to conduct large-scale clinical research using the methods identified in this study. In addition, by securing a strong normal control group for clarifying the characteristics of tissue and plasma, and applying various analysis methods to cfDNA, including ctDNA, it is expected that we will be able to get one step closer to cancer diagnosis using liquid biopsy.

## 4. Materials and Methods

### 4.1. 40 Healthy Donor Plasmas, 22 Pairs of GC Tissue and Blood, DNA Extraction

In this study, tissue and blood were randomly collected in pairs from 22 patients with cancer stages I to IV (IRB: CUH2022-8-012). For comparative study and normalization, 40 healthy human plasma samples were used (Innovative Research, Novi, MI, USA). cfDNA was collected using the Apostile cfDNA prep kit (Beckman Coulter, Brea, CA, USA). Genomic DNA was extracted from the tissue using a DNeasy kit (Qiagen, Hilden, Germany) and cut into sizes of less than 200 bp using Bioruptor Pico (Diagenode, Denville, NJ, USA). The size and quality of all nucleic acids was measured using TapeStation 4150 (Agilent Technologies, Santa Clara, CA, USA) and a Qubit 4 Fluorometer (Thermofisher Scientific, Waltham, MA, USA).

### 4.2. MeDIP Sequencing

For the MeDIP sequencing of cfDNA and fragmented gDNA, we followed previous methods [42]. The sequencing data were generated using a MGI G400 sequencer. Each output of sequencing produced more than two times the depth.

### 4.3. Data Processing and Mapping

The Cutadapt-2.4 program was used to select high-quality sequence data [43]. Adapter sequences and low-quality bases from the back of the original sequence were removed to obtain a high-quality sequence. The high-quality sequences were mapped to the hg38 human genome, and the mem command of the BWA-0.7.15 program was used for the mapping task [44]. Duplicate sequence data were removed using the MarkDuplicates command in the Picard-2.7.1 program. Flowcharts and tools used in data processing, such as DMR, CMR, and SFR, are available in Appendix A.

## Figures and Tables

**Figure 1 ijms-25-07377-f001:**
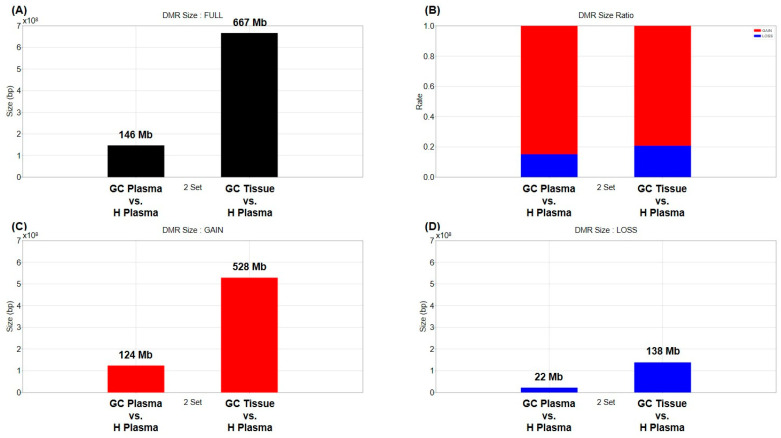
Size and composition of DMR obtained by comparing tissue and plasma of gastric cancer patients with healthy donor plasma. (**A**) The graph shows the total number and size of DMRs obtained by comparing the gastric cancer patients’ plasma or tissue to healthy donor plasma. (**B**) Red and blue bars represent the proportion of methylation gain and loss among total DMRs, respectively, when compared to heathy donor plasma. (**C**,**D**) The graph shows the total size of methylation gained and methylation lost.

**Figure 2 ijms-25-07377-f002:**
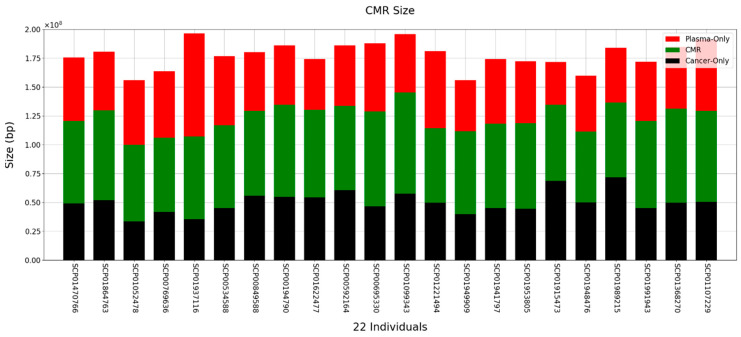
Size and distribution of CMR obtained by comparing tissue and plasma of gastric cancer patients. Red and black bars represent DMRs found only in the patient’s plasma and tissue, respectively. The green bars show commonly presented methylation regions in the patients’ plasma and tissue.

**Figure 3 ijms-25-07377-f003:**
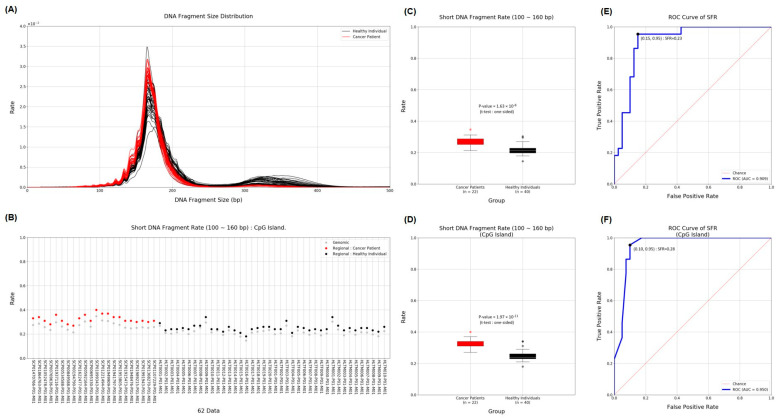
Distribution of SFR and ability to distinguish between gastric cancer and normal samples. (**A**) The graph shows the length and distribution of cfDNA obtained from the plasma of cancer patients (red) and healthy people (black). (**B**) SFRs were displayed for each individual and divided into genomic regions (grey) and CpG islands (red and black). (**C**,**D**) The graph shows the SFR between the two groups of cancer patients and healthy people in genomic and CpG island regions, respectively. (**E**,**F**) The graph shows the ROC curve and AUC score of SFR in genomic and CpG island regions, respectively. ROC, receiver operating characteristic; AUC, area under the ROC curve.

**Figure 4 ijms-25-07377-f004:**
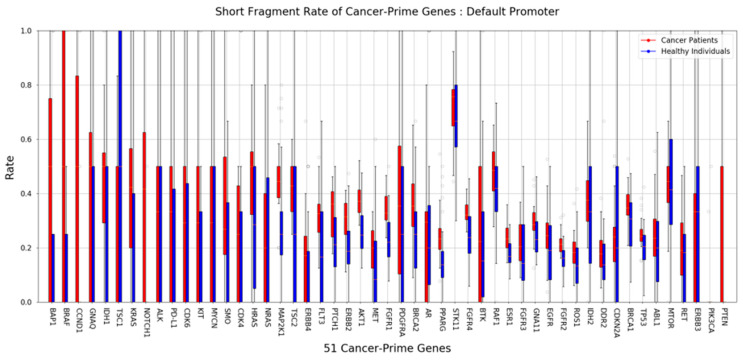
Differences in SFR within cancer-related genes. The graph shows the differences in SFR in the plasma of 22 patients (red) and 40 healthy people (blue) for each promoter of 51 cancer-related genes.

**Table 1 ijms-25-07377-t001:** Information of specimens in the study.

Group	Ethnicity	Tissue	Sample	Capture	# Samples	# Individual
Gastric Cancer	Korean	Plasma	cfDNA	MeDIP	22	22
		Cancer	gDNA	MeDIP	22	
Healthy	Hispanic	Plasma	cfDNA	MeDIP	40	40

**Table 2 ijms-25-07377-t002:** List of the top 10 significant 5′end-motifs.

Motif	Average Frequency (Rate)		*p*-Value
	Cancer Patients	Healthy Individuals	
CCAC	0.012198	0.010076	5.03 × 10^−11^
AGCT	0.003254	0.004846	6.11 × 10^−11^
TCCT	0.004531	0.003161	6.11 × 10^−11^
TCAG	0.002890	0.001785	8.16 × 10^−11^
TACG	0.000666	0.000449	9.85 × 10^−11^
AGTC	0.002373	0.003192	9.89 × 10^−11^
AGTT	0.002356	0.004342	1.20 × 10^−10^
TCAA	0.002575	0.001606	1.20 × 10^−10^
AGAT	0.002691	0.003770	1.32 × 10^−10^
AGCC	0.003252	0.003959	1.45 × 10^−10^

## Data Availability

All data are available upon reasonable request to the corresponding author of the article.

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
