# Peer review of "Detection and Characterization of Methylated Circulating Tumor DNA in Gastric Cancer"

_ijms, 2024, doi:10.3390/ijms25137377_

Round 1

Reviewer 1 Report

Comments and Suggestions for Authors

Agree to publish after minor revisions:

1. Please explain the reasons for the different ethnicities of the sample sources in the discussion section, and analyze whether this will have an impact on the analysis of the experimental results.

2. In 2.3 (In the Short Fragment Ratio ) section, there are many literature reports that the peak of cfDNA is around 167bp, while the peak of ctDNA is around 146bp. However, this article describes two different types of DNA as a distribution interval, with cfDNA in the range of 100 to 240 bp and ctDNA in the range of 100 to 160 bp. Is this differentiation feasible for experimental interpretation? Please explain in the discussion section.

Author Response

Thank you very much for taking the time to review this manuscript. Please check the attached file for more details.

Reviewer 2 Report

Comments and Suggestions for Authors

The study conducted by Seo et al. explored the diagnostic utility of methylated DNA fragments in gastric cancer. The authors used the MeDIP platform to collect methylated DNA fragments from tissue and plasma samples. They provided differentially methylated region (DMR), Commonly methylated region (CMR), and short fragment ratio (SFR) in gastric cancer. They also found 41.2% methylated fragment overlaps between tissues and paired plasma. I have the following questions about this study.

1.    The main limitation of this study is the healthy controls were from Hispanics (IRB?), and the sample size is small. It’s unclear about the age of cancer patients and healthy controls, as well as the stages of samples from gastric cancer. It’s important to know the status of the 20 individuals with gastric cancer (early stage or late stage? before treatment or after treatment?) if you are looking for circulating markers to diagnose gastric cancer.

2.    There is a description of “By analyzing these substances, it was expected that not only the presence or absence of cancer but also the type of cancer can be diagnosed”, but this study only focused on gastric cancer not other types of cancer. It’s of interest to check the BAP1 or those short sequences (4 bp) of fragments in other cancers.

3.    Figure 4 shows the differences in SFR within cancer-related genes, however, there are no SFR values for the PIK3CA gene (if it’s zero why you included it in Figure 4), and the SFR value seems higher in the healthy group for the TSC1 gene. Also, significant differences (e.g. BAP1) should be labeled in Figure 4.

4.    The font size for the X-axis and Y-axis in the figures can be adjusted. Missing Supplementary Table 1? Typos such as Line 119 “almost of CMRs…” .

5.    The statistical analysis in result 2.4 is inappropriate as you were comparing the frequency/proportion between the two groups. Also, Supplementary Figure 2 presents the DMR between tissues/plasma of patients and healthy controls, it’s of interest to add the DMR number between the tissues and plasma of patients.

6.    It’s still unclear which methylated cfDNA can be used as potential diagnostic biomarkers in this study and how it will be applied to the clinic.

Author Response

(The authors gave the same response as above.)

Round 2

Reviewer 2 Report

Comments and Suggestions for Authors

The authors responded to the comments and made revisions.

Only one comment needs to be further addressed: 

My confusion for Section 2.4 is about a description in lines 167-168 " The t- test was used in the statistical test for frequency differences between groups, and p-value adjustment was performed by the Benjamini-Hochberg procedure", to my knowledge, the t-test (determines difference between the means of two groups) is not suitable for comparing frequency/rates.  Table 2 is confusing, it seems that you were comparing frequency/rate between cancer patients and healthy individuals (difference between two groups) by using t-test? And if it's only two groups, what's the purpose of Benjamini-Hochberg adjustment?  This needs to be explained.

Author Response

Let's use the Wilcoxon rank-sum test instead of the T-test.

Benjamini-Hochberg adjustment is a comparison between two groups, but since p-values ​​are calculated for multiple motifs, the work was performed with the idea of ​​solving the problem of multiple-testing.

We will process it using the Wilcoxon rank-sum test and work together to achieve the Benjamini-Hochberg adjustment.

After analyis, we will remove and replace the methods in manuscript.